# Perceptions, Attitudes, and Knowledge toward Advance Directives: A Scoping Review

**DOI:** 10.3390/healthcare11202755

**Published:** 2023-10-18

**Authors:** João Carlos Macedo, Francisca Rego, Rui Nunes

**Affiliations:** 1Nursing School, University of Minho, Campus de Gualtar, 4710-057 Braga, Portugal; 2Health Sciences Research Unit: Nursing (UICISA: E), Nursing School of Coimbra (ESEnfC), Av. Bissaya Barreto, 3046-851 Coimbra, Portugal; 3Research Centre for Justice and Governance (JusGov), School of Law, University of Minho, Campus de Gualtar, 4710-057 Braga, Portugal; 4Faculty of Medicine, University of Porto, 4200-319 Porto, Portugal; mfrego@med.up.pt; 5Center of Bioethics, Faculty of Medicine, University of Porto, 4200-319 Porto, Portugal; ruinunes@med.up.pt

**Keywords:** advance directives, advance care planning, living will, perceptions, attitudes, knowledge

## Abstract

(1) Background: Advance directives are an expression of a person’s autonomy regarding end-of-life care. Several studies have shown that the level of completion in countries where advance directives are legalised is low. To better understand this phenomenon, it is important to know the perceptions, attitudes, and knowledge that the population has about this instrument. The aim of this article was to explore a population’s perceptions and/or attitudes and/or knowledge toward advance directives. (2) Methods: A search was conducted in March 2023 in the ISI Web of Knowledge, Scopus, and PubMed databases using the following keywords: “advance care directives”, “advance care planning”, “perceptions”, “attitudes”, and “knowledge”. Two hundred and twenty-four (224) articles were identified, and thirteen (13) were included for analysis. (3) Results: The selected articles point to a low level of knowledge toward advance directives: they recognise a strong positive attitude of the population toward the implementation of advance directives but a low level of achievement. (4) Conclusions: Studies on perceptions/attitudes/knowledge toward advance directives are important to understand the real needs of the population regarding this issue and to implement more adequate and effective promotion and dissemination measures.

## 1. Introduction

In recent years, with special emphasis since the beginning of the century, we have seen that the ethical principle of autonomy in decision making regarding health has been strengthened [1,2]. In other words, the right of the individual to exercise self-determination has been affirmed as a guarantee of respect for their dignity. In addition, this is so much so that it is no longer enough to exercise self-determination in the present, and we have reached the consensus—at least in some countries with mature democracies—that prospective autonomy must also be respected. We enter the realm of the wills that a person can leave written and that must be respected in the future. It is no longer a matter of obtaining the person’s consent for an act that will be conducted in the present but rather the formalisation of consent, an expressed will that will have its eventual manifestation of respect in the future in the case that the person is incapable of communicating. In short, we are discussing the advance directives (ADs), which are considered a milestone in civilisation for all the above reasons [3]. This document, which was born in the 1960s in the USA at the hands of a lawyer [4,5], is now an instrument that has spread throughout the world, with signifying the inalienable respect for the self-determination of the person that wrote this document [3,6].

This document was gradually disseminated around the world, with a greater impact in the 1990s in several US states, and in the following decades in European countries [1,6]. It was becoming admissible to respect the wishes of people who, for reasons of illness, were now incapable of expressing themselves, but who had written down their wishes regarding the healthcare they wished to refuse and/or receive in end-of-life situations. This form of respect for prospective autonomy, known as ADs, can take two distinct forms, which are not mutually exclusive [3,7,8]:

(a) Living will: a document in which the person expresses the healthcare they refuse or wish to receive if they are unable to express their wishes autonomously.

(b) Durable power of attorney for healthcare: a document which allows an individual to appoint a proxy to make healthcare decisions on their behalf when they lose the capacity to do so.

We know from numerous studies that the implementation of ADs has shown different rates of completion in different countries. In the United States of America, a 2008 Congress report indicated a completion rate between 18% and 36% of the population who had prepared ADs [9], and a more recent publication from 2017 states that approximately 1/3 of North Americans have completed ADs [10,11]. In Europe, on the other hand, studies point to a prevalence between 0.66% and 19% [12,13,14,15,16,17]. In Australia, there is a low prevalence [18] completion is at approximately 6% [19]; in Portugal, one of the few studies carried out in 2017 on the subject points to a figure of approximately 1.4% [20] and in a more recent one from 2021 a figure of 2.34% is reported [15], although the data from the Portuguese National Health Services point to the registration of approximately 53,000 ADs in January 2023, which corresponds to a value of 0.51% completion [21].

We believe that it is necessary to delve deeper into the study of the phenomenon of the low completion of ADs. We have found that some empirical studies focus on the perceptions of health professionals regarding ADs [22,23,24,25,26,27], the evaluation of the implementation of ADs [28,29,30,31,32,33,34,35], and the preferences and motivations of citizens to create ADs [36,37,38,39]. However, it seems important to us to understand how people perceive/understand and/or what attitudes they have toward ADs. In this sense, this scoping review was created to answer the following question. What are people’s perceptions, attitudes, and knowledge regarding ADs? Furthermore, in our review of the literature from the past five years, we found no other study that has specifically addressed this issue.

## 2. Materials and Methods

The main goal of the search was to explore studies on the perceptions and/or attitudes and/or knowledge of the population regarding ADs.

### 2.1. Search Strategy

This search took place in March 2023 in three databases, Web of Knowledge (ISI), Scopus, and PubMed, using the following MeSH terms: “advance directive” or “advance care planning” and “perceptions” and “attitudes” and “knowledge”. We limited the search period to the last five years (January 2018 to December 2022, inclusive). Table 1 displays the combination of terms used in each of the databases.

### 2.2. Inclusion Criteria

We included all empirical studies involving participants over 18 years of age from the general population.

### 2.3. Exclusion Criteria

Studies involving healthcare professionals or involving the family or health students were excluded. Editorials, letters, case studies, methodological studies, and literature reviews were also rejected.

## 3. Results

### 3.1. General Data

From the total of two hundred and twenty-four (224) articles identified in the databases, twenty-five (25) were removed for being duplicates. After analysing the title and abstract, one hundred and twenty-seven (127) articles were excluded. In the next phase, after reading the full text of seventy-two (72) articles assessed for eligibility, fifty-nine (59) records were excluded due to the following reasons: twenty-one (21) were literature reviews, thirty-seven (37) were research involving health professionals, family and/or health students, and one (1) was a letter to the editor. Finally, we reached thirteen articles to analyse. According to PRISMA guidelines [40], we prepared a flow diagram (Figure 1).

To better understand the results, we present Figure 2 which shows the country of origin and the number of selected articles.

The thirteen (13) selected articles are compiled in Table 2, which includes the following information: title of the article/author(s), country, and year of publication, participants in the study, objectives, type of study, and main outcomes.

Of the selected articles, 53.8% (*n* = 7) followed a quantitative methodology, 30.7% (*n* = 4) followed a qualitative methodology, and 15.3% (*n* = 2) were mixed studies.

Most of the articles 53.8% (*n* = 7) [41,42,43,44,45,46,47] described investigations in people with pathologies; 30.7% (*n* = 4) in the elderly and young adult population [12,48,49,50]; and 15.3% (*n* = 2) focused on the general population [15,51].

**Table 2 healthcare-11-02755-t002:** Mapping of the articles included in the scoping review.

Main Author/Year/Country	Participants	Objectives	Methodology/Study Type	Main Outcomes
Bar-Sela et al. (2021); Israel [41]	Advanced cancer patients (*n* = 109)	Evaluate the barriers and motives among Israeli cancer patients regarding advance care planning.	Mixed methods: cross-sectional, descriptive study	Participants mentioned that information and open communication were the main enabling factor to complete advance care planning. Communication with staff was rated more significant than with family members. The main motive to complete advance care planning was to ensure that the best medical decisions would be made and to avoid unnecessary medical procedures. Most of the participants did not hear about advance care planning from another source outside the hospital. Participants mentioned that the correct timing for implementing the ACP was during the terminal stage of the disease.
Cadmus et al. (2019); Nigeria [50]	A person aged 60 years and above (*n* = 34)	Explore the knowledge, attitude, and belief of older persons regarding decision making surrounding end-of-life life and advance directives.	Qualitative: exploratory study	The older person said they knew the term advance directives; however, when asked about care to receive or refuse, they could not answer. Most older persons preferred to have their children (first male son) as the major decision makers after their demise. Barriers to the implementation of advance directives were high legal fees and cultural rites and practices.
Carbonneau et al. (2018); Canada [42]	Patients with cirrhosis (*n* = 17)	Explore patients’ experiences and perceptions of the advance care planning (ACP) process in cirrhosis.	Qualitative: exploratory study	Participants expressed an overall lack of understanding of the role of advance care planning (ACP) processes. Most participants had a substitute decision maker. All participants mentioned the involvement of the family in the ACP process. Many saw ACP as critical to reducing the decision-making burden on the family. All participants agreed that discussions/conversations about ACP should happen outside of the hospital and not during acutely ill periods. Early anticipatory planning that requires discussion/conversation should be initiated in primary and outpatient care contexts.
Dhingra et al. (2020); USA [49]	Chinese American Immigrants older adults (*n* = 179)	Describe attitudes and beliefs concerning ACP in older, non-English-speaking Chinese Americans.	Quantitative: exploratory study	A total of 84.9% never completed an advance directive. A total of 56.8% were unfamiliar with any of the advance directives. A total of 74.4% were willing to complete one in the future. The rate of patients in ACP among Chinese immigrants is about half that of the general U.S. population.
Hou et al. (2021); China [43]	Patients with advanced cancer (*n* = 275)	Describe the knowledge and attitude of Chinese patients with advanced cancer toward advance care planning (ACP).	Quantitative: exploratory study	A total of 82.2% of patients had never heard about ACP. A total of 83.0% of patients had never talked about ACP. A total of 18.3% of patients were not willing to talk about ACP. A total of 67.8% of patients chose to refuse resuscitation attempts or life-sustaining medical interventions. A total of 70.8% of patients expressed a desire to have surrogate decision makers in the event they became unconscious, with their spouses being identified as the most significant proxy decision maker. Age, gender, place of residence, educational status, and family economic status were independent predictors of ACP.
Kim et al. (2018); Republic of Korea [44]	Older people with chronic diseases (*n* = 112)	Examine knowledge, attitudes, and barriers/benefits regarding advance directives (Ads) and their associations with AD treatment preferences among chronically ill, low-income, community-dwelling older people.	Quantitative: descriptive correlational study	A total of 8.9% of the participants knew about ADs. A total of 54.5% of the participants preferred hospice care. Few of the participants preferred aggressive treatments: 14.3% cardiopulmonary resuscitation (CPR), 9.8% ventilation support, and 8.9% haemodialysis. Being married was associated with the likelihood of preferring CPR and ventilation support. Higher education was associated with preferring the likelihood of CPR and haemodialysis. Having a cardiovascular disease/stroke was associated with the likelihood of preferring CPR and hospice care. Greater perceived barriers increased the likelihood of CPR preference and decreased the likelihood of hospice care. Greater perceived benefits decreased the likelihood of CPR and ventilation support. Advance directives knowledge decreased the likelihood of haemodialysis preference.
Kleiner et al. (2019); Switzerland [12]	Older adults aged ≥71 (*n* = 1701)	Test the hypothesis of an association between increased knowledge of ACP and a more positive perception.	Quantitative: descriptive, correlational study	A total of 47% of the participants were aware of the legal dispositions for ACP. A total of 14% of the participants had completed or were in the process of completing an AD. There is a positive association between the knowledge of ACP and a more positive perception of ADs.
Laranjeira et al. (2021); Portugal [15]	Adults (aged ≥18 years) (*n* = 1028)	Assess the knowledge, attitudes, and preferences of a sample of Portuguese adults regarding end-of-life care decisions and advance care directives.	Quantitative: descriptive, correlational study	A total of 26.63% of the participants were unaware of what an advance care directive (ACD) was. A total of 2.4% of the participants had an ACD. Higher levels of knowledge were associated with more positive attitudes.
Lim et al. (2022); Malaysia [51]	Adults (*n* = 385)	To assess the knowledge, attitude, and practice of community-dwelling adults and their associated factors.	Quantitative: cross-sectional, descriptive study	A total of 5.2% of the participants were aware of ACP. A total of 85.7% of the participants had a positive attitude toward ACP. A total of 84.4% of the participants felt that ACP was necessary and would consider discussing an ACP.
Schnur et al. (2019); EUA [48]	Young adults aged 18–26 years (*n* = 147)	Identify associations among young adults’ characteristics, knowledge of ACP, and readiness to engage ACP-related behaviours.	Quantitative: cross-sectional, descriptive, correlational study	A total of 93.2% of the participants reported thinking that good quality of life was more important than living as long as possible. A total of 84.7% of the participants scored positive toward ACP. A total of 78.9% of the participants scored a positive disposition toward the ACP process. Less than 4% of the participants reported engaging in ACP-related conversations with their doctors or healthcare providers. Higher ACP knowledge scores were weakly associated with more positive views of ACP.
Sprange et al. (2019); Canada [45]	Cirrhosis patients (*n* = 97)	Assess knowledge and recall of participation in ACP.	Mixed: exploratory study	A total of 33% of the participants had completed a personal directive (PD). A total of 14% of the participants had completed a goals of care designation (GCD). A total of 78% of the participants believed that GDC is important. A total of 84.5% of the participants preferred initiating the ACP discussion in an outpatients’ clinic setting. The participants considered specific qualities during the ACP discussion: - Good communication skills. - Patient empathy. - Ability to educate. - Medical expertise. - Frank description of health outcomes and prognosis.
Ugalde et al. (2018); Australia [46]	Cancer patients (*n* = 14)	Explore the comprehension of ACP in people with cancer who have current advance care plans.	Qualitative: exploratory, descriptive study	Most participants demonstrated partial comprehension of their advance care plan. Participants’ attitudes and their written documents’ congruence were limited. Most participants reported creating an ACP was helpful because of the following reasons: - enabling them to work through what they want; - enabling them to feel empowered, a peace of mind, calmer, and relieved that their wishes are known. Some participants considered ACP advantageous because it removed difficult decision making away from their family and the doctors.
Wang et al. (2021); China [47]	Participants were patients with brain tumours who were older than 18 years and were reported. (*n* = 316)	Describe the knowledge and preferences of ADs and end-of-life care decisions of patients with tumorous.	Qualitative: cross-sectional, correlational study	A total of 88.61% of the participants had never heard of ADs. A total of 65.18% of the participants reported that they would like to make an AD. For those who would like to make an AD, the primary reasons were as follows: Ensure comfort at the end of life. Reduce financial burdens on their family. For those who would not like to make ADs, the primary reason was as follows: Lack of familiarity with the concept of ADs. Belief that doctors or family members would make decisions for them. A total of 79.43% of the participants wanted to discuss end-of-life arrangements with medical staff. A total of 63.29% of the participants were willing to receive end-of-life care, even though it would not delay death. Knowledge of ADs, receiving surgery or radiotherapy, age lower than 70 years, male sex, educational qualification of college and beyond, without children, medical insurance for nonworking or working urban residents, and self-payment of medical expenses were significant predictors of preferring to make ADs.

### 3.2. Perceptions, Attitudes, and Knowledge

In general, perceptions of ADs are positive. However, it should be noted that in countries without a defined legal framework for AD participants had lower perceptions of the subject in their studies [43,47,50,51]. Some of the studies revealed that between 82.2% and 86.6% of the participants had never heard of ADs [43,47], and there was also a study in which only 5.2% of the sample showed awareness of ADs [51]. In contrast, in some studies conducted in countries where there are legal regulations about ADs, awareness is higher: 47% in Switzerland [12], 56.8% in the USA [49], and 76.3% in Portugal [15].

In addition to perceptions, or the lack of them, the articles revealed that peoples’ level of knowledge about ADs is low [12,15,44,45,46,47,49,51]. The values referred to in the investigations about the level of knowledge ranged from 8.9% [44] to 26.63% [15] of the samples under study. Notwithstanding these data, participants in some studies stated that ADs are important to ensure the exercise of autonomy and to avoid unnecessary medical procedures [41,46], to ensure comfort at the end of life and reduce the family’s financial effort [47], to empower the person, bring peace of mind, and ease the family and physicians’ decision making at the end of life of patients [46].

Regarding attitudes toward ADs, studies have shown that they are positive about both their use and application [15,47,48,51]. Proof of this is the fact that one of the studies had positive attitude scores of 65.1% [47] and another of 85.7% [51].

### 3.3. AD Completion

In terms of AD completion, there is a variability of values. However, it can be observed that completion rates are higher in samples in patients with diseases [41,42,43,44,45,46,47] and lower in studies that examined the healthy population [12,15,49].

In percentage terms, one study showed a value of 2.4% for AD completion [15] and another of 33% [45].

### 3.4. Correlation Factors and Predictors

Another important fact that has been highlighted by some studies is the fact that there is a correlation between the variables of knowledge and attitudes, i.e., high levels of knowledge about ADs were associated with more positive attitudes [15,48], or more positive perceptions [12], and even the refusal of more aggressive/invasive treatments [44].

In terms of the development of ADs, studies have shown some predictors. Thus, in one of the studies with cancer patients, the variables of age, place of residence, level of education, and economic status were identified as the predictors of AD completion [43]. In another study in patients with a brain tumour, factors including knowledge, treatment (surgery or radiotherapy), age (over 70 years), gender (male), higher education level, not having children, and having health insurance were identified as predictors for performing ADs [47].

### 3.5. Communication and Planning

A result that emerged from some studies was the importance and preference given to communication from health professionals to help prepare ADs [47]. Despite this finding, one study reported that less than 4% of the young adult sample had discussed ADs with a health professional [48].

Furthermore, in some studies in patients (with cancer and liver cirrhosis), in relation to the process of preparing an AD, participants mentioned that the preparation of an AD should be planned outside the hospital environment/system [42,45], and mentioned that the preparation of an AD should not occur in periods when the disease is aggravated [41,42]. On the other hand, in one of the studies with patients with diseases, participants pointed out some characteristics that they considered very important in the process of advising health professionals about ADs: teaching and communication skills, empathy, medical knowledge, and honesty in the transmission of the diagnosis and prognosis [45].

## 4. Discussion

This reveals, across all studies, that the population’s knowledge about ADs is extremely low. Despite some cultural and clinical specificities of some study samples, the empirical data reveal a population with an important level of illiteracy about ADs. This fact has been maintained over time, as already noted in a systematic literature review conducted between 1994 and 2016 on the population’s knowledge about ADs [52], as well as other studies [13,14,16,53]. This is perhaps one of the most important data points to highlight because of its practical consequence, i.e., despite the existence of this instrument for the exercise of prospective autonomy, a citizen’s right, its residual knowledge contributes to the low rates in the implementation of ADs, as found by the studies in this review and comparable with other evidence [17,20,52]. As we mentioned before, these data have also been maintained over time, with the USA showing higher rates, and reporting that one-third of the population will have ADs [10,11]. Moreover, the rates are lower in Europe, and are between 0.66% and 19% [12,13,14,15,16,17]. There may be several reasons for this discrepancy, despite the cultural differences between populations, but it is necessary to note that in the USA, the discussion and legalisation of ADs began in the 1970s, while in most countries, especially European, discussion on this issue began only at the beginning of the 21st century [54,55]. Regardless of this time lag, which may justify some progress in the numbers in the USA, it seems to us (in line with the evidence of this review and corroborated by other studies) that the knowledge deficit is a barrier to not starting the process of the elaboration of ADs [17,52].

The lack of knowledge about ADs is high, and as we were able to see from the studies in countries without a legal framework for ADs that the samples in the reviews exhibited a lack of knowledge about ADs, with values exceeding 80%. Even so, in countries with a legal framework, the rates of lack of knowledge were 56.8% [49] and 47% [12], demonstrating that even with the implementation of a law, measures must be taken to increase the literacy of the population in this context. It seems to us that there is a need to involve political decision makers in the area of health and, simultaneously, involve health professionals so that projects can be developed in the community to provide citizens with more information and training to prepare ADs [52,56,57].

Even so, we found that the attitudes and perceptions about ADs are very favourable, which indicates that, despite the populations’ low levels of knowledge in this area, when they are informed about what an AD entails, they provide a very positive assessment and display a growing interest in the subject. In some studies, participants have the following perceptions about the added value of ADs: respect for the person’s autonomy, avoiding therapeutic futility, ensuring comfort at the end of life, and relieving the family and physicians from ethical decision making in end-of-life care. These are in the same line of evidence pointed out by other studies [11,52,57].

An important fact about communication in end-of-life care planning and the role of health professionals should be highlighted. In line with the results of the review conducted, other studies confirm the importance of individuals discussing the care they receive or decline at the end of life. For the participants in the research studies, this conversation would be important and more effective for elaborating ADs if they had the support of a healthcare professional [58,59,60,61,62,63]. It seems evident to us that a citizen, when faced with doubts which are admittedly technical, should want to obtain advice. Incidentally, this idea is already in practice in the USA under the Federal Patient Self-Determination Act of 1990, and health institutions are obliged to inform citizens of their right to prepare their AD and to provide advice to this effect [64,65]. In summary, with this scoping review, we can state that despite the geographical variety of studies and the different legal frameworks on ADs, we were able to briefly identify the following results: there are positive attitudes/perceptions and extremely poor knowledge and understanding of ADs.

### 4.1. Limitations

Although we agree with the idea that a scoping review in this area can provide added value in broadening the knowledge in this field of health literacy and bioethics, in particular [66], we are aware that it has some limitations. First, as the search is in only three databases, it may miss some important studies that are not referenced (for example, a search in a law database may provide other data). On the other hand, many of the selected studies do not present statistically equivalent samples to the national populations, which limits the generalisation of the data. Additionally, the limitation of the research to the last five years allows us to keep the data up to date; however, the temporal analysis of the evolution of the data is more compromised. Finally, we believe that the time during which this review was performed was mostly focused on the pandemic period of COVID-19 and this may have an effect of the data and results, which may have a decreasing effect on scientific production.

### 4.2. Implications for Practice and Research

Despite the limitations, we believe that this review provides valuable insights for both practice and further research on this subject. Thus, we believe that further research is needed on the knowledge on ADs, particularly with samples that closely resemble the populations of different countries. This approach will enable us to obtain more reliable data on the current state of the field in question. On the other hand, there is a need to create projects in the area of health education, involving health professionals—as connoisseurs of more technical matters—to empower citizens in the preparation of ADs [20,58,67,68]. In other words, the attitudes and perceptions of the population about ADs are positive; therefore, it is essential to promote dialogue and open, technically adequate communication, equipping citizens with the knowledge they need to make informed decisions about preparing ADs for end-of-life care planning. Understanding the population’s attitudes toward ADs, as well as their knowledge and attitudes, is crucial for enhancing healthcare practices and increasing completion rates. This should always be performed while considering a person’s autonomy and best interests.

## 5. Conclusions

An advance directive (AD) is a tool that enables the exercise of prospective autonomy. It serves as instructions for the care a person wishes to receive or refuses in the event they become incapable of expressing their preferences at the end of life. This is what we could call a reinvention of the practice of prospective informed consent [69]. This instrument is currently widespread in several countries; however, the level of completion of the population to prepare informed consent is low. Understanding how the population perceives, knows, and what attitudes it has toward ADs will allow us to identify the needs in this area, and to direct more appropriate and stimulating measures. This review, by mapping the studies on the population’s perceptions/attitudes/knowledge of ADs, contributes to a better understanding of the problems that the population faces when they want to plan their end-of-life care. This review found that the samples of the population under study showed positive attitudes and perceptions toward ADs. However, the level of knowledge on the subject is low due to certain cultural and clinical differences, and low values of AD completion are also present.

There is still a need for further research in this area, especially to obtain more evidence using research tools to assess levels of knowledge and identify real problems that that citizens have. There is a need to increase evidence in this area to enable us to respond more adequately with measures that may help a citizen to exercise his/her right to prospective autonomy.

## Figures and Tables

**Figure 1 healthcare-11-02755-f001:**
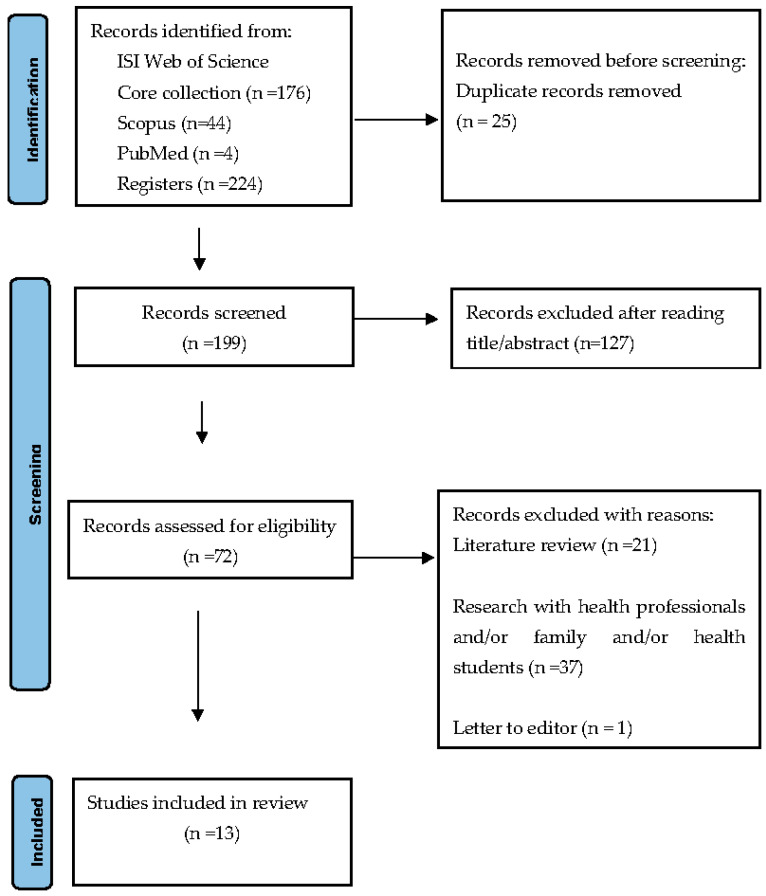
PRISMA flow diagram (adapted).

**Figure 2 healthcare-11-02755-f002:**
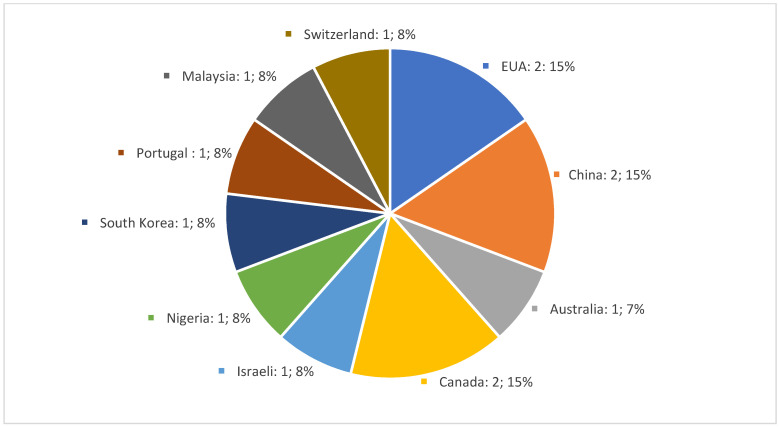
Country of origin of articles/number and %.

**Table 1 healthcare-11-02755-t001:** Search strategies.

Search Equation	Database	Results
Advance directives or advance care planning or living will (topic) and perception (topic) and attitudes (topic) and knowledge (topic)	WoS * (core collection)	176
(TITLE-ABS-KEY (advance directives) or TITLE-ABS-KEY (advance care planning) or TITLE-ABS-KEY (living will) and TITLE-ABS-KEY (perception) and TITLE-ABS-KEY (attitude) and TITLE-ABS-KEY (knowledge))	Scopus	44
(((Advance directive [title/abstract]) or (advance care planning [title/abstract])) or (living will [title/abstract])) and (perception [title/abstract])) and (attitude [title/abstract])) and (knowledge [title/abstract]	PubMed	4

* WoS: Web of Science.

## Data Availability

Not applicable.

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
