# Peer review of "Perceptions, Attitudes, and Knowledge toward Advance Directives: A Scoping Review"

_healthcare, 2023, doi:10.3390/healthcare11202755_

Round 1

Reviewer 1 Report

Overall

Thank you for the opportunity to read this manuscript. This manuscript provides a lot of information on the perceptions surrounding advanced directives.

This manuscript’s insight regarding knowledge, perceptions and attitudes surrounding advanced directives could be very enlightening for researchers investigating the effects of advanced directives in the space today.

I do have a few comments which I believe could strengthen the argument and the clarity of the manuscript.

Abstract

Abstract does a good job of introducing the review, what ADs are, and why there is a need to study them, as well as what was found in the relationships studied.

Introduction

p.2, paragraph between lines 49 and 59. Is “adherence” the right word in this paragraph? The way that I am reading the manuscript makes it sound like these are the rates of having an AD within the different regions. When I hear “adherence”, I’m not thinking about how many of something an area has, but instead are they adhering to what was said or prescribed. So to me, the phrase advanced directive adherence reads as being adherent to an advanced directive when it is used. I would clarify if that is the best word for what is being discussed or if there is another way to phrase that sentiment.

Materials and Methods

p.3, Line 80. Is this supposed to be “all empirical studies with subjects over 18 years of age”? If so, that makes more sense.

Results

p.3, lines 89 and 91. “Several reasons” and “with reasons” tripped me up some. It made sense when looking at the figure below. I think dropping the “for several reasons” in line 89 would be fine, then specify in line 91 “excluded for the following reasons”.

The table is a very good way to display the included selections and the information provided clearly illustrates what was seen by researchers.

P. 11, line 174: This feels like something is missing before AD. Based on my understanding, these are predictors of having an advanced directive. Since an AD is something you would have in place, I feel like that is what is being measure, not simply the AD itself.

Discussion

For the limitations, I would like to know what a law database was not used, given that the manuscript discusses the history of advanced directives and how they arose out of the legal space instead of the healthcare space. Would more have been found there?

There are a few questions of grammar and word choice for clarity above, but overall very good work. 

Author Response

Thank you

Reviewer 2 Report

I think the article needs to go for English editing 

The introduction is short; you need to expand it a little more. 

Consider elaborating on the criteria used for selecting the studies included in the review, Such as 

  • Did you include the studies that use only the English language?
  • Publication time of the included studies (Did you use only the studies published in the last five years?).
  • Did you limit your review to a journal article or include another publication type, such as a thesis?  
  1. The gap in knowledge needs to be explained further ( what your studies (review) will add to the previous ones. 
  2. It is essential to explain further why the exclusion studies were excluded from your studies, and it will be better if you give examples of these studies

I think the article needs to go for English editing 

Author Response

Thank you.
